# Timing of Dietary Fatty Acids to Optimize Reduced Risk of Type 2 Diabetes Mellitus: Findings from China Health and Nutrition Survey

**DOI:** 10.3390/nu17132089

**Published:** 2025-06-24

**Authors:** Hao Ye, Yuqi Wu, Pan Zhuang, Xiaohui Liu, Yang Ao, Yin Li, Jianxin Yao, Haoyin Liu, Zongmei Yang, Yu Zhang, Jingjing Jiao

**Affiliations:** 1Department of Endocrinology, The Second Affiliated Hospital, School of Public Health, Zhejiang University School of Medicine, Hangzhou 310012, China; yh1999@zju.edu.cn (H.Y.); liuxiaohui@zju.edu.cn (X.L.); aoyang9@zju.edu.cn (Y.A.); liyin-@zju.edu.cn (Y.L.); haoyinliu@zju.edu.cn (H.L.); zongmeiyang@zju.edu.cn (Z.Y.); 2Department of Gastroenterology, The First Affiliated Hospital, Zhejiang University School of Medicine, Hangzhou 310003, China; 22013038@zju.edu.cn (Y.W.); panzhuang@zju.edu.cn (P.Z.); yaojianxin@zju.edu.cn (J.Y.); y_zhang@zju.edu.cn (Y.Z.); 3Department of Food Science and Nutrition, College of Biosystems Engineering and Food Science, Zhejiang University, Hangzhou 310058, China

**Keywords:** Chrononutrition, type 2 diabetes mellitus, dietary fatty acid timing, n-3 PUFAs, plant-sourced MUFAs

## Abstract

Background/Objectives: Chrononutrition highlights the significance of temporal consumption behavior for a healthy dietary pattern. This study investigated the relationship between dietary fatty acid (FA) intake timing and type 2 diabetes mellitus (T2DM) risk. Methods: A total of 14,518 participants in the China Health and Nutrition Survey (1991–2015) were recruited. Dietary intake and mealtime were collected via three consecutive 24 h dietary recalls. Multivariable Cox proportional hazard models were employed to estimate the association between FA intake at meals and T2DM risk. Differences in FA intake between dinner and breakfast (Δ = dinner-breakfast) were calculated for each type of FA intake. Sensitivity analyses considering the effects of snacks, the sum of different types of FAs, and other confounding factors were performed. The isocaloric substitution model was used to view the risk changes according to the shifted mealtime. Results: During an average of 10.1-year follow-up, 1048 T2DM cases occurred. T2DM risk was inversely associated with n-3 polyunsaturated FAs (n-3 PUFAs) (*p* trend = 0.032) and plant-sourced monounsaturated FAs (P-MUFAs) (*p* trend = 0.002) intake at dinner versus breakfast. The highest category of Δ n-3 PUFAs and Δ P-MUFAs were respectively linked to a 19% and 29% reduced T2DM risk. Unanimous associations were found for the difference between lunch and breakfast rather than dinner and lunch. Isocalorically switching 1 standard deviation (SD) of n-3 PUFAs or MUFAs intake at breakfast with the corresponding type of FAs at dinner was associated with a 22% and 20% lower risk of T2DM, respectively. Conclusions: Timely consumption of dietary FAs, particularly n-3 PUFAs and plant-based MUFAs at dinner versus breakfast, is crucial for T2DM prevention.

## 1. Introduction

The booming diabetic population around the world, especially in China, imposes a considerable burden on life and socio-economic development. Given the close link between circadian disruption and metabolic disturbance, many risk factors, including the mistiming of meals, shift work, energy overloading, and fat consumption at night, have been identified as increasing the risk of type 2 diabetes mellitus (T2DM) [1,2]. This warrants scientific research to find potential approaches based on chronobiology for the prevention and management of T2DM; dietary intervention would be the cornerstone in this field. Previous studies have found that restricted time eating, intermittent eating, and optimal dietary patterns at dinner were effective in controlling glucose metabolism and fluctuation by regulating the function of the circadian system [3,4,5]. However, large prospective population investigations were limited, and it was hard to clarify the extent to which the dietary components distributed at each meal impact the incidence of T2DM.

Nutrition epidemiology consistently suggested focusing on the quality and quantity of dietary fatty acids (FAs) for metabolic health and T2DM prevention [6,7,8]. In our previous research, marine n-3 polyunsaturated FAs (PUFAs) consumption elevated the risk of T2DM in a dose-dependent way [9], while monounsaturated FAs (MUFAs) intake showed no significant association [10]. However, some researchers held different opinions from ours [11,12]. A recent study found that the short-term substitution of saturated fat with unsaturated fat does not significantly affect insulin sensitivity or β-cell function [11]. Another study based on 16,290 adults from the National Health and Nutrition Examination Survey found that total MUFA and PUFA intake may be essential in preventing prediabetes and T2DM [12].

The heterogeneity of previous studies may stem from differences in study types, populations, dietary investigation methods, and the time of diet intake, etc. Temporal eating patterns and nutrients, also known as Chrononutrition, bring new insight into the effect of timed dietary food intake on circadian rhythm synchronization involved in maintaining metabolic homeostasis [13,14]. Research on Chrononutrition has suggested that time of day is indicative of having an influence on the postprandial glucose response to a meal, which, therefore, has a major effect on type 2 diabetes [13]. This emerging field highlights that certain food components should be aligned to periods of the day according to the internal circadian system to optimize the health benefits of nutrients [14]. Chrononutrition may be conducive to explaining the heterogeneous results regarding the relationship between fatty acid intake and T2DM risk, as reported in the current literature.

To better understand the effects of different subtypes of fatty acid consumption timing on T2DM development, we aimed to assess the differences in dietary fatty acid consumption between dinner and breakfast and their association with T2DM risk and also to compare the differences with lunch, using data from the China Health and Nutrition Survey (CHNS). Grounded in Chrononutrition, the current study prioritizes temporal redistribution effects over absolute intake, which shows a reduced T2DM risk when shifting FAs to dinner. The Δ metric captures circadian-driven metabolic variations, offering actionable insights into meal-specific FA timing for T2DM prevention.

## 2. Materials and Methods

### 2.1. Study Population

Data of this study was utilized from the CHNS, a multistage nationwide cohort, which was approved by institutional review committees at the University of North Carolina, Chapel Hill, and the National Institute for Nutrition and Health, Chinese Center for Disease Control and Prevention (CCDC). The project was conducted from the year of 1989 to 2015 to comprehensively evaluate the nutrition status and health-related outcomes during the societal and economic transition in China. A total of 10 waves (1989, 1991, 1993, 1997, 2000, 2004, 2006, 2009, 2011, and 2015) of data were collected in this survey. All participants signed informed consent. The descriptions of CHNS have been expounded in detail elsewhere [15,16]. In short, the survey was conducted in 9 provinces with randomly selected subjects in both urban and rural areas [15]; three other autonomous cities were not added until 2011 [17].

A total of 22,399 individuals who participated in at least two surveys from 1991 to 2015 were included in the study initially. Then, we first excluded participants younger than 20 years of age at entry (*n* = 6306) and those without complete dietary information (*n* = 204). Persons were further excluded due to previously diagnosed T2DM (*n* = 717) or cardiovascular disease (*n* = 331) and inconvincible energy intake (out of range of 800–4200 kcal/d for men and 600–3500 kcal/d for women; *n* = 293). Finally, a total of 14,518 subjects were included in the current study (Figure 1).

### 2.2. Dietary Assessment and Main Exposure

During the survey, face-to-face interviews based on structured protocol were conducted by well-trained staff to ensure the uniformity and accuracy of data collection. From the beginning to the end of each 3d home inventory, household food consumption was calculated by carefully scaling and recording all food consumed (including cooking oils, salt, and other condiments). In Chinese cuisine, most meats and vegetables were basically cooked with these oils so that energy and dietary fatty acids from cooking oils at separate times for each subject were estimated according to the proportion of specific oil-added foods the individual consumed [18]. The household survey section plus individuals’ food consumption at home or away from home were assessed via three consecutive 24 h recalls, which have been validated for assessing dietary intakes [16]. For the calculation of energy, macronutrients (carbohydrate, protein, and fat), dietary cholesterol, and subtypes of fatty acids from consumed foods, versions of the Chinese Food Composition Table were referred to, including the 1991 version used for the data from 1997 and 2000 survey, and the 2002 and 2004 versions [19,20,21], which were incorporated to evaluate nutritional components.

Total fatty acid intake (in grams) was determined by multiplying the weight of each food item (in grams) by their respective fat content (in grams per 100 grams), and subsequently applying the appropriate conversion factors. Next, specific fatty acid intakes were determined by multiplying the total fatty acid intake by the respective percentage composition of each fatty acid type. The intake of dietary fatty acids includes saturated fatty acids (SFAs), MUFAs, and PUFAs, classified according to the number of unsaturated bonds. In order to investigate more specifically, we classified MUFAs into plant-derived MUFAs (P-MUFAs) and animal-derived MUFAs (A-MUFA) according to different food sources. P-MUFAs are derived from plant-based foods, such as vegetables, rice, fruits, legumes, and nuts, including palmitoleic acid (PA) and oleic acid (OA). A-MUFAs are extracted from animal foods, including animal oil, red meat, eggs, poultry, fish, and dairy products [10]. PUFAs were further divided into n-3 series and n-6 series according to the position of the unsaturated bonds. Dietary n-3 PUFAs derived from fish, shrimp, crab, shellfish, and nuts, including α-linolenic acid (ALA), eicosapentaenoic acid (EPA), and docosahexaenoic acid (DHA). The dietary sources of n-6 PUFAs include soybeans, nuts, poultry, and eggs, and n-6 PUFAs include arachidonic acid (AA) and linoleic acid (LA). To reflect a long-term diet, we used the cumulative average of repeatedly measured FA intakes in each wave.

Each dietary questionnaire also contained at which mealtime various food intake. For mealtime, six options, breakfast, morning snack, lunch, afternoon snack, dinner, and evening snack, could be selected for the corresponding food item. Then, separate energy, macronutrients, and subtypes of dietary fatty acid intake at each meal could be assessed and were expressed as a percentage of energy (% en) at separate mealtimes [1,22]. The accumulative means of nutrient intake from each wave were calculated to ensure a robust habitual diet with minimized variation.

The exposure variables were the differences in dietary fatty acid intake timing, expressed with label “Δ”, indicating the difference in the consumption of fatty acids at dinner versus breakfast (main exposure), dinner versus lunch, or lunch versus breakfast.

### 2.3. T2DM Ascertainment

T2DM status was ascertained based on the American Diabetes Association criteria for the diagnosis of diabetes and the guideline for the prevention and treatment of type 2 diabetes mellitus in China, according to any of the following criteria: (1) reported a diagnosed T2DM through questionnaire inquiring; (2) showed high fasting plasma glucose over 7.0 mmol/L or high HbA1c over 6.5% in the 2009 survey; (3) received diabetes treatments (the special diet, weight control, oral medication, insulin and Chinese traditional medicine). A previous study based on the 2009 survey confirmed the accuracy of diagnosed diabetes in this cohort [18,23,24].

### 2.4. Confounding Measurements

In addition to the total intake of energy (kcal/day), protein (% en), and carbohydrate (% en), we also calculated total dietary cholesterol intake (% en) and cereal consumption (g/day). Considering the potential association of dietary quality with T2DM [25], we used 7 dietary components to score an Alternative Healthy Eating Index (AHEI) based on the AHEI-2010 standard and scoring criteria [26]. In brief, higher consumption of vegetables, fruit, cereal fibers, legumes, and nuts, as well as less consumption of sugar-sweetened beverages (SSB), red and processed meats, and sodium, indicate better dietary quality. Each diet component scored from a minimum of 0 (worst) to a maximum of 10 (healthiest), and then the sum of each part was presented as AHEI, ranging from 0 to 70 in total. A higher AHEI score indicated that the participant had a higher quality diet. All the above covariates were categorized into quintiles.

Other confounding measurements covering sociodemographic and lifestyle factors were recorded during the survey as well. Age, gender (male or female), ethnic (Han or non-Han), body mass index (BMI), marital status (never married, married or living as married, widowed/divorced/separated, or unknown), education (illiteracy, <high school, or ≥high school), income (quintile), and urbanization index (quintile), physical activity (no regular activity or light activity, moderate activity, or vigorous activity), smoking (non-smoker, ever-smoker, current smoker), alcohol use (yes or no), and history of hypertension (yes or no), were included in as nondietary data. BMI was calculated by dividing body weight by the square of body height (kg/m^2^).

### 2.5. Statistical Analysis

Continuous variables were described as mean ± standard error (SE), and discrete variables were described as percentages of the subjects in each category of the differences in dietary n-3 PUFAs or P-MUFAs intake at dinner versus breakfast (Δ = dinner − breakfast) by quintiles. For baseline characteristics comparisons, 2 tests and general linear models were conducted for continuous variables and categorical variables, respectively. Both the total and Δ intake of three macronutrients were expressed in the dietary profile. Person-years of follow-up were calculated from the year of baseline questionnaire to the earliest of the following: the year when participants self-reported physician-diagnosed incident diabetes, confirmed by a fasting blood test (2009), death, or the final wave of follow-up (2015).

Cox proportional hazard regression models were used to compute hazard ratios (HRs) and 95% confidence intervals (CIs) for dietary fatty acid intake timing (the difference in amount between each meal) and T2DM in a time-dependent manner. Covariates, including age, gender, ethnicity, BMI, marital status, education, income, urbanization index, physical activity, smoking, drinking, diagnosed hypertension, total intake of energy, protein, carbohydrate, dietary cholesterol, and cereal intake, remaining meal and remaining subtypes of dietary fatty acids were controlled in multivariable-adjusted models. Then, linear trends across categories of Δ were examined by generalized linear models.

We also performed substitution analysis to estimate the effects of theoretically replacing one type of dietary fatty acid intake in the forenoon with one in the evening. In the equivalent substitution analysis, the consumption of total energy remained constant based on Cox hazards models. The standard deviation (SD) standardization method was used to process the data regarding dietary fatty acid intake during the forenoon and evening (including main meal and snack) in order to eliminate the effect of unit dimension. The processed data conformed to the standard normal distribution with a mean value of 0 and an SD value of 1 [27].

Four sets of sensitivity analyses were conducted. First, we combined breakfast and morning snacks in the forenoon and combined dinner and evening snacks in the evening, then reanalyzed the difference in a similar way to examine the change in the risk of T2DM. Next, considering the positive results may be affected by the total fatty acid intake, where the health benefits like the consumptions of P-MUFAs and n-3 PUFAs in total may gloss over the effects of timing, the ratio of Δ/sum was calculated in the second set of sensitivity analysis. We further excluded individuals with extreme BMI (<18.5 or >40 kg/m^2^) to see whether the findings remained unchanged. Finally, we additionally adjusted for AHEI based on model 3. To investigate how the disparity in FA intake between dinner and breakfast influences the risk of T2DM among individuals of varying ages, we performed a subgroup analysis stratified by age, using 40 years as the cutoff point. This age threshold was selected because early-onset diabetes is defined as diabetes diagnosed prior to the age of 40 [28].

Statistical analyses were performed by version 9.4 SAS Statistical Analysis Software. 95% CIs of HRs were presented, and a two-sided *p* value < 0.05 was considered significant after Bonferroni correction for multiple comparisons in Cox proportional hazards models.

## 3. Results

### 3.1. Population Characteristic

Among 14,518 participants included in this study, a total of 1048 T2DM cases were identified after an average of 10.1 follow-up years. Baseline characteristics were summarized according to the differences (Δ) in dietary n-3 PUFAs (dinner vs. breakfast) or P-MUFAs (dinner vs. breakfast) intake in quintiles, as shown in Table 1 and Appendix A. For Δ n-3 PUFAs, age, ethnicity, BMI, annual income, education level, physical activity, history of hypertension, and most dietary factors, except for total dietary energy and cholesterol intake, were significantly different among groups. Individuals with higher consumption of n-3 PUFAs at dinner tended to consume fewer carbohydrates and cereals but more fat, protein, MUFAs, and SFAs at dinner and were less likely to reach a higher AHEI score. More interestingly, the lowest amount of total n-3 PUFA intake was not subject to the Q1 group, which showed a “U” shape across quintiles. Similar trends could be seen in terms of quintiles of Δ plant-sourced MUFAs intake.

### 3.2. Associations of Dietary Fatty Acid Intake Timing with the Risk of T2DM

The associations of the differences in multiple dietary fatty acid consumption between dinner and breakfast with T2DM risk were presented in Table 2, including the crude model. In the fully adjusted model 3 with additional adjustment of dietary confounding, the highest quintile of Δ n-3 PUFAs consumption was associated with a 19% reduced risk of T2DM compared with participants in quintile 1 (*p* trend = 0.032). Although no documented association was found for Δ n-6 PUFAs intake, higher n-6/n-3 PUFAs consumed at dinner over breakfast were more likely to elevate T2DM risk (HR 1.38, 95% CI 1.12–1.71, *p* trend = 0.009). Additionally, among different degrees of unsaturation of dietary fatty acids, only Δ MUFAs showed a significant inverse association with T2DM risk with a *p* trend = 0.041, probably due to the effect of plant-sourced MUFAs distributed at dinner and breakfast (Appendix A). With increasing categories of Δ P-MUFAs consumption, the difference in timed intake of MUFAs from plant sources was related to 15%, 15%, 21%, and 29% reductions in risk of T2DM (*p* trend = 0.002).

Dietary fatty acid consumption at lunch is also of interest. Similarly, the results shown in Figure 2 suggested that higher intake of n-3 PUFAs and plant-based MUFAs at lunch rather than breakfast were significantly associated with a lower risk of T2DM (*p* trend = 0.023 for Δ n-3 PUFAs, *p* trend < 0.001 for Δ P-MUFAs), while the mealtimes of dinner and lunch did not serve a potential effect.

### 3.3. Substitution Analyses

In substitution analyses, we observed altered risks of T2DM when the equivalent certain fatty acid consumption was switched from forenoon to evening per SD. Specifically, the risk of T2DM could be reduced upon the isocaloric replacement of each SD of dietary n-3 PUFAs (HR = 0.78, 95% CI: 0.65–0.94) or MUFAs (HR = 0.80, 95% CI: 0.69–0.93) consumed in the forenoon with that in evening correspondingly, while remained unchanged by the substitution with the other type of dietary fatty acids. Additionally, there was a slight association between substitution per SD dietary P-MUFAs in the forenoon with evening and T2DM risk (Table 3).

### 3.4. Associations of Subtypes of Dietary Fatty Acid Intake Timing with the Risk of T2DM

Further analysis of subtypes of dietary n-3 PUFAs found that Δ ALA and Δ DHA consumption between dinner and breakfast was inversely associated with a reduced risk of T2DM (Appendix A). The highest quintile of Δ ALA consumption between dinner and breakfast was associated with a 17% reduced risk of T2DM compared with participants in quintile 1, while the highest quintile of Δ DHA consumption was associated with a 56% reduced T2DM risk compared with those in quintile 1. However, no documented significant association was found for Δ EPA intake with T2DM. For subtypes of P-MUFAs, Δ PA and Δ OA intake were not found to be significantly associated with T2DM risk.

### 3.5. Sensitivity Analysis

Firstly, considering the snack consumption of dinner and breakfast, the findings did not change substantially, in which a larger difference between n-3 PUFA or P-MUFA intake in the forenoon and evening was significantly associated with a lower risk of T2DM (Appendix A). Similarly, all of the above findings remained unchanged when the sum of corresponding fatty acid intake was considered (Appendix A). In the third set, after excluding individuals with extreme BMI (*n* = 2218), those out of 18.5 to 40 kg/m^2^, the observed association of n-6/n-3 intake at dinner versus breakfast with the risk of T2DM was weakened with no statistical significance, while for the consumptions of n-3 PUFAs and plant-based MUFAs, the results remained significant (Appendix A). In the final set, additional adjustments for dietary AHEI score did not appreciably change the results (Appendix A).

### 3.6. Subgroup Analysis

Significant interactions were found between age with Δ n-6/n-3 PUFAs and Δ P-MUFA intake between dinner and breakfast (*p* interaction < 0.05, Appendix A). Further analysis found that the harmful influence of Δ n-6/n-3 PUFAs intake between dinner and breakfast was more pronounced among individuals aged more than 40 years old (HR_Q5 vs. Q1_ 1.45, 95% CI 1.13–1.87, *p* trend = 0.012), as well as the beneficial impact of P-MUFA (HR_Q5 vs. Q1_ 0.70, 95% CI 0.55–0.90, *p* trend = 0.010).

## 4. Discussion

This nationwide, long-term cohort study first assessed the associations of the differences in various dietary fatty acid intake timing with the development of T2DM in China. Our findings indicated that a reduced risk of T2DM related to a higher intake of n-3 PUFAs and P-MUFAs at dinner rather than breakfast, similar to the substitution results, which filled the gap in Chrononutrition for T2DM prevention.

Previous research demonstrated the role of MUFAs in maintaining glycolipid homeostasis [29,30], which supported our findings. According to a Japanese cohort study including 19,088 individuals, higher intake of MUFA was inversely associated with T2DM [6]. However, some studies showed that dietary MUFA has no protective effect on T2DM risk [31]. The inconsistency among these studies may be ascribed to the failure to differentiate the source of MUFAs. In China, MUFAs are mainly consumed from animal-derived fats (e.g., lard and tallow) and pork, followed by plant-derived oils (e.g., peanut oil and canola oil), rice, and nuts [32]. Therefore, it is important to consider the source of food providing MUFAs for the Chinese population [33]. We previously reported that MUFA intake from plant-based sources, instead of animal-sourced, could largely increase the risk of T2DM in China [10]. Consistently, our previous study also indicated that intakes of MUFAs from plant but not animal sources were associated with lower total mortality [33]. However, a subtype study found that the intake of PA and LA between dinner and breakfast has no protective effect on the risk of T2DM, indicating that the protection of P-MUFA for T2DM depends more on the total amount. The current study added further detail to the timing of MUFA intake: the prevention of T2DM development is related to higher consumption of MUFA, especially P-MUFA, at dinner or lunch than breakfast.

General dietary guideline highly recommends more PUFA intake in daily meals for diabetic patients owning to the cardioprotective effect [34,35]. Nevertheless, the relationship between dietary PUFA intake and the incidence of T2DM was subjected to debate, even showing an elevated T2DM risk upon a higher intake of DHA [8]. It could be explained by ethnic background, where total PUFA consumption related to reduced risk of T2DM in Asia (RR = 0.897, 95% CI: 0.860–0.936) out of other regions [35]. For the treatment of diabetic patients, both marine n-3 PUFAs and plant-sourced n-3 PUFAs significantly elicited insulin secretion after the 6-month intervention, indicating the potential of n-3 PUFAs in glucose control [36]. Meanwhile, the types and sources of PUFAs and the extent of their distribution in three meals should also be considered. Our research emphasized that n-3 PUFA consumption at dinner or lunch would be better than breakfast. Intriguingly, Oishi et al. [37] have previously dug out the health benefits of fish oil in a time-dependent way in a mouse model. Administrating fish oil in the morning significantly increased circulating DHA and EPA levels and improved lipid metabolism when compared with feeding time at night [37]. The inconsistent results may be a consequence of the nocturnal nature of mice. More time-dependent nutritional intervention studies are of importance in supporting this hypothesis.

To the best of our knowledge, this is the first study to examine the association of fatty acid distribution across meals throughout the day with T2DM. The most important finding of this study was that a higher intake of n-3 PUFA and P-MUFA at dinner than breakfast was significantly associated with lower T2DM risk, and this association was independent of a series of traditional dietary risk factors, in particular, snack intake and diet quality. Our study suggests that not only nutritional values but also the timing of meals need to be taken into consideration for dietary recommendations for patients with diabetes.

Given the reciprocal connections between circadian disruption and diabetes, increasing studies emphasized the therapeutic strategy of resetting circadian clocks, more importantly, through dietary intervention. Macronutrients are key regulators of peripheral clocks and could coordinate or desynchronize internal circadian clocks [38]. Therefore, dietary fatty acids may directly affect both the circadian clocks of the hypothalamus and peripheral tissues through the blood–brain barrier to stimulate fatty acid sensing signals and molecular clocks [38,39]. The expression and activity of glucokinase (GCK), which plays a key role in the onset of diabetes, are affected by the circadian rhythm and are directly related to the metabolic disorders of T2DM. Studies have shown that the expression and activity of GCK exhibit circadian rhythm fluctuations and are directly controlled by Circadian Locomotor Output Cycles Kaput (CLOCK) and aryl hydrocarbon receptor nuclear translocator-like protein 1 (BMAL1). BMAL1 binds to the E-box in the promoter region of GCK and is a clock-controlled gene. The disruption of these circadian rhythm genes may lead to abnormal GCK function, thereby affecting glucose metabolism. In addition to circadian rhythm regulation, the expression of GCK is also regulated by dietary signals. GCK is mainly regulated by dietary signals through the insulin-mediated sterol regulatory element-binding protein 1c (SREBP1c) pathway and direct regulation of glucose. Postprandial insulin activates SREBP1c and enhances GCK transcription, while elevated glucose levels directly upregulate GCK expression [40].

Previous studies have shown that DHA has a significant protective effect on the circadian rhythm control mechanism. In the presence of DHA, the destructive effect of palmitate on the expression of the *BMAL1* gene is reduced, indicating that DHA has a protective effect on the rhythmic system [41]. Existing studies implied that the protective effect of the minor allele of the CLOCK gene on insulin sensitivity was related to dietary MUFA intake. Regarding n-3 PUFAs, recent Chrononutrition science has delineated its direct relationship with the circadian clock system, which is involved in the regulation of the biological clock and metabolic rhythm [38,42]. Fish oil, rich in n-3 PUFA, was also reported to have a phase-shifting effect on the liver clock. Additionally, fish oil can increase glucose concentration by secreting GLP-1 through the GPR120 receptor in the large intestine and establish the synchronization mechanism of the biological clock by enhancing insulin secretion, suggesting food containing fish oil or DHA/EPA is ideal for adjusting the peripheral clock [42]. Moreover, Maresin 1, one of the derivates of DHA, was proven to facilitate circadian clocks as the natural ligand binding to circadian clock receptor RORα [39]. However, the underlying mechanism of functional fatty acids consumed within different time windows on circadian clock coordination and glycaemic control was still uncovered, which further researches are supposed to delve into.

This study has some strengths, including its prospective design for long observation periods, accurate T2DM diagnosis, and precise calculation of macronutrients and multiple fatty acid intake in a single meal (cumulative averages), which has never been examined before. We additionally employed sensitivity analyses for considerations of snacks across three meals, the sum of different types of fatty acid consumption, dietary patterns, etc., to verify the reliability of our findings.

Several limitations in this research should also be noticed. First, the consumption of classified dietary fatty acids at each mealtime was probably underestimated even though the mean value during each round based on the 24 h dietary recall method was well evaluated [15]. Second, dietary fatty acids derived from household cooking oils were proportionately measured by roughly evaluating the amount of specific oil-added foods for each household member. In this sense, cooking methods and eating behaviors vary among families; for example, certain foods may be cooked with or without oils, which may result in bias. Third, a lack of information about chronotype, linked with the biological clock, which differs across individuals, potentially influenced the results. Also, these findings may not be generalized to other populations. Fourth, the effects of residual confounding, especially complicated dietary factors, could not be covered, although additional adjustments of many reported covariates related to T2DM and diet quality in sensitivity analyses. Fifth, although the CHNS study examined the consumption of diverse foods and cooking methods, due to the inability to precisely evaluate the alterations in fatty acid levels post-cooking, we could not explore the influence of different fatty acids in foods prepared by different cooking methods on T2DM development risk. Finally, we could not investigate the effect of marine n-3 PUFAs or specific dietary fatty acid intake timing due to the relatively low consumption for Chinese people, let alone the distribution at each meal. However, it is necessary to design precise intervention studies in terms of the eating windows to gain solid evidence. The underlying effect of timed dietary fatty acid intake merits further research.

## 5. Conclusions

The optimized protective effects of dietary n-3 PUFAs and plant-sourced MUFAs on the T2DM risk were found when increasing the consumption at dinner versus breakfast, suggesting more attention on fat intake timing. With the increasing population of T2DM and the role of circadian clocks, more precise nutritional research is warranted to move forward in a time-dependent manner.

## Figures and Tables

**Figure 1 nutrients-17-02089-f001:**
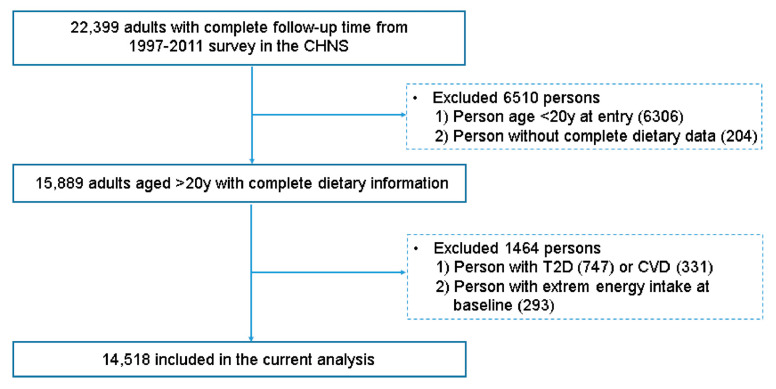
Flow chart of the participants in the China Health and Nutrition Survey (1991–2015).

**Figure 2 nutrients-17-02089-f002:**
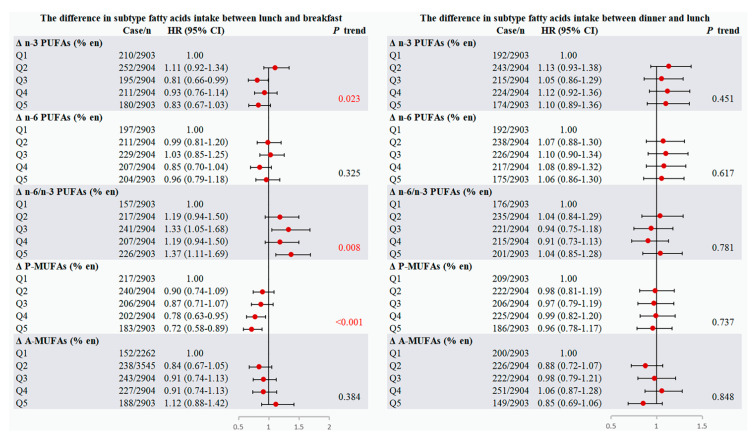
Multivariable-adjusted HRs (95% CIs) of T2DM in terms of dietary fatty acid intake timing at lunch in the China Health and Nutrition Survey (1991−2015) (*n* = 14,518). The association of the differences in subtypes of fatty acid intake between lunch and breakfast (**left**) or the differences in subtypes of fatty acid intake between dinner and lunch (**right**) with the risk of T2DM. HRs (95% CIs) were estimated using time-dependent Cox proportional hazards models with adjustment for age, gender, ethnicity, BMI, education, marital status, income, urbanization index, physical activity, smoking, drinking, history of hypertension, total energy intake, dietary cholesterol, cereal, carbohydrate and protein intake, dinner/breakfast FAs intake and remaining subtypes of dietary FAs.

**Table 1 nutrients-17-02089-t001:** Baseline characteristics of subjects in the China Health and Nutrition Survey across categories of the difference in dietary n-3 PUFAs intake between dinner and breakfast (n = 14,518) †.

Characteristics	Quintiles of n-3 PUFAs Between Dinner and Breakfast					*p* Value
	Q1 ‡	Q2	Q3	Q4	Q5	
Δ § n-3 PUFAs (% en)	−1.64 ± 0.678	−0.04 ± 0.001	0.12 ± 0.001	0.30 ± 0.001	0.77 ± 0.010	<0.001
*n*	2903	2904	2904	2904	2903	
Age (years)	43.1 ± 0.3	43.3 ± 0.3	43.7 ± 0.3	43.3 ± 0.3	45.4 ± 0.3	<0.001
Male (%)	47.1	47.8	46.9	48.1	45.2	0.212
Han (%)	91.8	90.4	87.5	85.4	88.4	<0.001
Body mass index (kg/m^2^)	23.5 ± 0.1	22.9 ± 0.1	22.8 ± 0.1	22.5 ± 0.1	23.0 ± 0.1	<0.001
Household income (yuan/yr)	34,277.3 ± 708.4	24,899.6 ± 592.9	26,407.3 ± 619.3	27,732.5 ± 654.4	35,058.3 ± 842.6	0.045
Urbanization index	68.0 ± 0.4	54.9 ± 0.4	57.5 ± 0.4	59.5 ± 0.4	66.9 ± 0.4	0.060
Married (%)	85.7	84.6	85.0	85.1	84.2	0.589
≥middle school (%)	36.5	22.5	22.0	23.3	28.8	<0.001
Vigorous activity (%)	22.6	41.3	40.3	39.2	27.4	<0.001
Current smoker (%)	29.1	30.4	29.6	30.2	28.8	0.054
Alcohol drinker (%)	37.2	34.5	33.0	35.3	35.0	0.060
History of hypertension (%)	14.5	15.7	13.4	12.6	15.2	0.003
Dietary intake						
Total energy (kcal/day)	2048.1 ± 10.3	2169 ± 10.2	2161.1 ± 9.9	2151.0 ± 9.2	2068.6 ± 10.3	0.468
Δ Energy	230.4 ± 5.5	174.6 ± 4.7	258.1 ± 4.9	322.3 ± 5	302.7 ± 5.6	<0.001
Total carbohydrates (% en)	55.0 ± 0.2	61.1 ± 0.2	59.5 ± 0.2	57.6 ± 0.2	52.3 ± 0.2	<0.001
Δ Carbohydrates (% en)	0.3 ± 0.3	−3.5 ± 0.2	−8.3 ± 0.2	−14.6 ± 0.2	−19.1 ± 0.3	<0.001
Total protein (% en)	13.4 ± 0.1	12.7 ± 0	12.8 ± 0	13.3 ± 0.1	14.0 ± 0.1	<0.001
Δ Protein (% en)	1.9 ± 0.1	1.3 ± 0.1	2.2 ± 0.1	3.5 ± 0.1	4.7 ± 0.1	<0.001
Total fat (% en)	31.6 ± 0.2	26.3 ± 0.2	27.6 ± 0.2	29.1 ± 0.2	33.7 ± 0.2	<0.001
Δ Fat (% en)	−2.4 ± 0.3	2.6 ± 0.2	7.2 ± 0.2	12.6 ± 0.2	16.4 ± 0.3	<0.001
Total cereal (mg/day)	375.0 ± 2.5	430.9 ± 2.7	411.7 ± 2.4	398 ± 2.4	355.8 ± 2.3	<0.001
Total cholesterol (mg/day)	0.3 ± 0.004	0.3 ± 0.005	0.3 ± 0.004	0.3 ± 0.004	0.3 ± 0.004	0.329
Total n-3 PUFAs (% en)	1.2 ± 0.01	0.8 ± 0.01	0.7 ± 0.01	0.8 ± 0.01	1.3 ± 0.01	<0.001
Total n-6 PUFAs (% en)	8.9 ± 0.1	7.4 ± 0.1	7.3 ± 0.1	7.0 ± 0.1	8.0 ± 0.1	<0.001
Total SFAs (% en)	8.1 ± 0.1	7.0 ± 0.1	7.8 ± 0.1	8.5 ± 0.1	9.7 ± 0.1	<0.001
Total MUFAs (% en)	13.4 ± 0.1	11.0 ± 0.1	12.1 ± 0.1	13.0 ± 0.1	15.7 ± 0.1	<0.001
AHEI ¶	39.5 ± 0.1	41.1 ± 0.1	39.7 ± 0.1	38.6 ± 0.1	37.8 ± 0.1	<0.001

† Data are means ± SE unless otherwise indicated. Individual income was inflated to 2015. Total energy intake was presented as energy density (g·2000 kcal^−1^·d^−1^), and individual fatty acid intake timing was expressed as the energy percentage of each separate meal. ‡ Q = quintile. § Δ = the difference in dietary components intake between dinner and breakfast (dinner − breakfast). ¶ AHEI = Alternative Healthy Eating Index.

**Table 2 nutrients-17-02089-t002:** Multivariable-adjusted HRs (95% CIs) for the association of the differences in subtypes of fatty acid intake between dinner and breakfast with the risk of T2DM in the China Health and Nutrition Survey (1991–2015) (*n* = 14,518).

	Quintiles of Dietary Intake of Fatty Acids					*p* Trend
	Q1	Q2	Q3	Q4	Q5	
Δ n-3 PUFAs intake between dinner and breakfast						
Δ, range (% en)	<−0.15	−0.15–0.04	0.04–0.20	0.20–0.41	>0.41	
Cases/*n*	233/2903	242/2904	188/2904	206/2904	179/2903	
Person-years	25,522	30,762	32,012	31,772	26,304	
Model 1	1.00	0.79 (0.66–0.95)	0.58 (0.48–0.70)	0.64 (0.53–0.77)	0.67 (0.55–0.81)	<0.001
Model 2	1.00	0.91 (0.76–1.10)	0.70 (0.57–0.85)	0.82 (0.68–0.997)	0.74 (0.61–0.90)	0.002
Model 3	1.00	0.94 (0.78–1.13)	0.72 (0.59–0.88)	0.88 (0.72–1.07)	0.81 (0.66–1.01)	0.032
Δ n-6 PUFAs intake between dinner and breakfast						
Δ, range (% en)	<−0.65	−0.65–0.70	0.70–1.95	1.95–3.48	>3.48	
Cases/*n*	222/2903	215/2904	208/2904	207/2904	196/2903	
Person-years	25,857	31,032	32,375	31,761	25,347	
Model 1	1.00	0.78 (0.65–0.94)	0.71 (0.59–0.86)	0.72 (0.60–0.87)	0.86 (0.71–1.04)	0.064
Model 2	1.00	0.85 (0.71–1.03)	0.80 (0.66–0.97)	0.82 (0.68–0.995)	0.91 (0.75–1.10)	0.248
Model 3	1.00	0.85 (0.70–1.03)	0.80 (0.66–0.97)	0.84 (0.69–1.03)	0.90 (0.74–1.10)	0.288
Δ n-6/n-3 PUFAs intake between dinner and breakfast						
Δ, range (% en)	<−2.91	−2.91–−0.18	−0.18–0.68	0.68–2.41	>2.41	
Cases/*n*	153/2903	227/2904	221/2904	227/2904	220/2903	
Person-years	28,781	29,545	31,269	30,514	26,263	
Model 1	1.00	1.42 (1.16–1.75)	1.35 (1.10–1.66)	1.45 (1.18–1.78)	1.73 (1.41–2.13)	<0.001
Model 2	1.00	1.24 (1.01–1.52)	1.17 (0.95–1.44)	1.27 (1.04–1.57)	1.38 (1.12–1.71)	0.005
Model 3	1.00	1.36 (1.07–1.73)	1.31 (1.02–1.68)	1.41 (1.11–1.79)	1.38 (1.12–1.71)	0.009
Δ P-MUFAs intake between dinner and breakfast						
Δ, range (% en)	<−2.21	−2.21–−0.26	−0.26–0.68	0.68–2.24	>2.24	
Cases/*n*	230/2903	224/2904	206/2904	210/2904	178/2903	
Person-years	24,200	30,454	31,312	31,944	28,462	
Model 1	1.00	0.71 (0.59–0.85)	0.63 (0.52–0.76)	0.62 (0.52–0.75)	0.57 (0.47–0.70)	<0.001
Model 2	1.00	0.84 (0.69–1.01)	0.82 (0.67–0.998)	0.79 (0.65–0.95)	0.71 (0.58–0.87)	0.001
Model 3	1.00	0.85 (0.70–1.03)	0.85 (0.70–1.05)	0.79 (0.65–0.97)	0.71 (0.57–0.88)	0.002
Δ A-MUFAs intake between dinner and breakfast						
Δ, range (% en)	<0.00	0.00–1.12	1.12–4.25	4.25–8.29	>8.29	
Cases/*n*	165/2366	275/3441	218/2904	219/2904	171/2903	
Person-years	21,454	37,475	31,095	31,560	24,788	
Model 1	1.00	0.96 (0.79–1.16)	0.91 (0.75–1.12)	0.90 (0.73–1.10)	0.94 (0.76–1.16)	0.394
Model 2	1.00	1.06 (0.87–1.29)	0.96 (0.78–1.18)	0.91 (0.74–1.12)	1.02 (0.82–1.27)	0.540
Model 3	1.00	1.06 (0.86–1.31)	0.96 (0.78–1.19)	0.95 (0.77–1.18)	1.09 (0.86–1.39)	0.918

HR, hazard ratio; CI, confidence interval; T2DM, type 2 diabetes mellitus; Δ, the difference in FAs intake between dinner and breakfast (dinner − breakfast); % en, percentage of energy at each meal; cases/n, the ratio of case to total. Model 1: Adjusted for age and gender (male or female). Model 2: further adjusted for ethnic (Han or non-Han), BMI (in kg/m^2^; <18.5, 18.5–23.9, 24–27.9, or ≥28), marital status (never married, married or living as married, widowed/divorced/separated, or unknown), education (illiteracy, <high school, or ≥high school), income (quartile), and urbanization index (quintile), physical activity (no regular activity or light activity, moderate activity, or vigorous activity), smoking (non-smoker, ever-smoker, current smoker), alcohol use (yes or no), history of hypertension (yes or no). Model 3: further adjusted for total energy intake (quintile), cholesterol intake (quintile), cereal intake (quintile), carbohydrate intake (% en, quintile), protein intake (% en, quintile), lunch FAs intake (% en, quintile) and remaining subtypes of dietary fatty acids (% en, quintile).

**Table 3 nutrients-17-02089-t003:** Multivariable-adjusted HRs (95% CIs) of T2DM: isocaloric substitution of different types of fatty acids from evening to forenoon (including main meal and snack).

	HR (95% CI)	*p* Value
Substitution per SD dietary MUFAs in forenoon with		
each per SD dietary MUFAs in evening	0.80 (0.69–0.93)	0.004
each per SD dietary SFAs in evening	0.89 (0.78–1.01)	0.080
each per SD dietary PUFAs in evening	1.00 (0.89–1.13)	0.983
Substitution per SD dietary plant-sourced MUFAs in forenoon with		
each per SD dietary plant-sourced MUFAs in evening	0.91 (0.82–1.02)	0.099
each per SD dietary animal-sourced MUFAs in evening	0.95 (0.84–1.07)	0.392
Substitution per SD dietary n-3 PUFAs in forenoon with		
each per SD dietary n-3 PUFAs in evening	0.78 (0.65–0.94)	0.009
each per SD dietary n-6 PUFAs in evening	1.04 (0.91–1.20)	0.532

HRs (95% CIs) were estimated using time-dependent Cox proportional hazards models with adjustment for age, gender, ethnicity, BMI, education, marital status, income, urbanization index, physical activity, smoking, drinking, history of hypertension, total energy intake, dietary cholesterol, cereal, carbohydrate and protein intake, lunch FAs intake and remaining subtypes of dietary FAs.

## Data Availability

The data are available from the CHNS on request (https://www.cpc.unc.edu/projects/china (accessed on 1 March 2025)).

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
