# Peer review of "Timing of Dietary Fatty Acids to Optimize Reduced Risk of Type 2 Diabetes Mellitus: Findings from China Health and Nutrition Survey"

_nutrients, 2025, doi:10.3390/nu17132089_

Round 1
Reviewer 1 Report
Comments and Suggestions for Authors
This paper investigates the dietary fatty acid intake and mealtime by three consecutive 24-h dietary recalls related to risk of T2D in 14 518 randomly selected Chinese participants in rural and urban areas approximately every second year during 1991-2015. Previous studies have shown different results relating fatty acid intake and risk of T2D. However, ref 9 showed that only fried fish showed an association to T2D, which might be related to which oil was used in cooking. In the present study dietary recording were made at home and the sum of the dietary components were scored to an Alternative Healthy Eating Index (AHEI) indicating the quality of the diet and categorized into quintiles but the authors are not giving any indicator how the n-3 was reported in relation to cooking. Confounding measurements were included the analyses, e.g. other dietary regimens and social factors. In this study 1048 cases with T2D were identified after an average of 10 follow-up years, and T2D risk was found inversely associated with n-3 polyunsaturated fatty acids and even stronger with plant-sourced monounsaturated fatty acids at dinner vs breakfast with a 19% and 29% reduced T2D risk, respectively. There were less differences between breakfast and lunch. Isocalorically switching between breakfast and dinner modified the risk. Interestingly the intake of both n-3 PUFA and plant-sourced MUFA showed an U-shape across quintiles for the difference between dinner and breakfast, which might be of interest to comment on in regard to social differences between different countries. Higher n-6/n-3 fatty acid intake was associated with significant higher risk of T2D but plant-sourced monounsaturated fatty acids and total amount of n-3 fatty acids expressed as difference between dinner and breakfast were associated with lower risk of T2D. Similar pattern was seen comparing lunch to breakfast but not comparing dinner and lunch. Substitution per SD of dietary MUFA or dietary n-3 PUFA in forenoon per SD of respective fatty acids in evening showed significant lower risk of T2D. The associations were similar when hazard ratios were calculated adjusted for different social and clinical factors, including snack consumptions.
The study elucidates the importance of fat quality in metabolism, shown in many animal studies. It would be of interest if the authors defined which MUFA, n-6 and n-3 fatty acids are included in the study, especially since other studies have shown there are large differences between the impact of linoleic acid and arachidonic acid as well as between the very long-chain n-3 fatty acids.
A-MUFA is not defined and therefore the difference to plant-sourced MUFA is lacking.
Reference list is incongruent with some journals in capital letters.
Ref 4 is incomplete.
Ref 25 and 35 are incomplete.
Ref 19-21 are incomplete
Author Response
Comments 1: In the present study dietary recording were made at home and the sum of the dietary components were scored to an Alternative Healthy Eating Index (AHEI) indicating the quality of the diet and categorized into quintiles but the authors are not giving any indicator how the n-3 was reported in relation to cooking.
Response 1: Thank you for pointing this out. In the current analysis, food intakes were assessed via three consecutive 24-h recalls and nutrition calculations were based on the corresponding versions of the Chinese Food Composition Table (FCT), where the nutrient levels of each food were recorded in detail. The total intakes of FAs were determined by multiplying the weight of each food by its respective fat content and then applying an appropriate conversion factor. Next, the specific intakes of FAs were determined by multiplying the total intake of FAs by the percentage composition of each type of fatty acid. We have clarified the FAs intake assessment in the Method section, which can be found in the revised manuscript at Lines 125-142, Page 4 in red.
Additionally, it is well-recognized that different cooking methods significantly influence the fatty acid profiles of foods [1, 2]. However, due to the lack of precise data on the intake of various fatty acids after cooking, it was unable to explore the impact of the intake of various fatty acids related to cooking on the risk of T2DM in the current study. We have added this limitation in the Discussion section, which can be found in the revised manuscript at Lines 421-425, Pages 12-13 in red.
References:
- Neff, M.R.; Bhavsar, S.P.; Braekevelt, E.; Arts, M.T. Effects of different cooking methods on fatty acid profiles in four freshwater fishes from the Laurentian Great Lakes region. Food Chem. 2014, 164, 544-550.
- Cui, Y.; Hao, P.; Liu, B.; Meng, X. Effect of traditional Chinese cooking methods on fatty acid profiles of vegetable oils. Food Chem. 2017, 233, 77-84.
Comments 2: It would be of interest if the authors defined which MUFA, n-6 and n-3 fatty acids are included in the study, especially since other studies have shown there are large differences between the impact of linoleic acid and arachidonic acid as well as between the very long-chain n-3 fatty acids.
Response 2: Thank you for pointing this out. Dietary MUFAs include palmitoleic acid (PA) and oleic acid (OA), n-6 PUFAs include arachidonic acid (AA) and linoleic acid (LA), and n-3 PUFAs include alpha - linolenic acid (ALA), eicosapentaenoic acid (EPA) and docosahexaenoic acid (DHA). We have further analyzed subtypes of diet FAs intakes between dinner and breakfast, and found that the highest quintile of Δ ALA and Δ DHA consumption between dinner and breakfast was associated with a 17% and 56% reduced risk of T2DM compared with participants in quintile 1, respectively. However, Δ EPA, Δ PA and Δ OA consumption between dinner and breakfast were found had no significant association with T2DM risk. The added analysis can be found in the revised manuscript at Lines 294-302, Page 10 and Table S3 in red.
Comments 3: A-MUFA is not defined and therefore the difference to plant-sourced MUFA is lacking.
Response 3: We apologize for not clearly explained the definitions of various fatty acids. P-MUFAs are obtained from plant-based dietary sources, encompassing vegetables, rice, fruits, legumes, and nuts, while A-MUFAs are sourced from animal-based foods, including various types of animal oil, red meat, eggs, poultry, fish, and dairy products. We have added a more detailed explanation in the method section, which can be found in the revised manuscript at Lines 125-136, Page 4 in red.
Comments 4: Reference list is in congruent with some journals in capital letters. Ref 4 is incomplete. Ref 25 and 35 are incomplete. Ref 19-21 are incomplete.
Response 4: We apologize for using the reference format incorrectly, and we have now corrected the reference format in the revised manuscript in red. For Ref 19-21, since we used the entire food composition table for nutrient calculation, no specific chapters or page numbers were marked.
Reviewer 2 Report
Comments and Suggestions for Authors
The authors present a quite interesting dietary study with a huge inclusion of participants (14,518) that has a lot of merit. The topic of the study is interesting, since I believe is the first time that the consumption of fatty acids at different mealtimes is analyzed to relate it to Type 2 diabetes risk. Some questions are raised to understand better the study, and I believe it will increase the quality of the manuscript.
- The dietary analysis was performed in the same individuals in all the waves? That is, it is a longitudinal dietary study? Or the volunteers were different in each wave of data collection, which would be part of a cross-sectional study.
- It is not clear how the diagnosis of T2D was managed. For example, if a diagnosis of T2D was detected in the 4th wave, how the dietary data was analyzed. Moreover, the time of diagnosis of T2D (in age) and fatty acid intake could also be interesting. Does the correct timing of fatty acids intake prevent the onset of T2D?
- Since the reduction of the risk of T2D is between 19 and 29%, why the authors concluded that the timing is crucial?
- I don't understand why the difference in the intake is used as an outcome and not the actual intake in each mealtime?
- Could you please explain the sources of omega-3 fatty acids in the studied population.
- Finally, since dietary habits change during life due to several factors, are there fluctuations in fatty acids intake in the whole period studied
Author Response
Comments 1: The dietary analysis was performed in the same individuals in all the waves? That is, it is a longitudinal dietary study? Or the volunteers were different in each wave of data collection, which would be part of a cross-sectional study.
Response 1: Thank you for your question. The current study is a longitudinal diet study. A total of 10 survey waves were conducted from 1991 to 2015. The participants in this study participated in at least 2 of the 10 survey waves. Meanwhile, we excluded the participants with baseline diabetes to ensure that the dietary survey was earlier than the diagnosis of T2DM. We have clarified in more detail in the Method section, which can be found in the revised manuscript at Lines 98-99, Page 3 in red.
Comments 2: It is not clear how the diagnosis of T2D was managed. For example, if a diagnosis of T2D was detected in the 4th wave, how the dietary data was analyzed. Moreover, the time of diagnosis of T2D (in age) and fatty acid intake could also be interesting. Does the correct timing of fatty acids intake prevent the onset of T2D?
Response 2: Thank you for your question. The Ascertainment of T2DM was mentioned in the Method section at Lines 153-161, Page 4, including self-reported T2DM, high fasting plasma glucose/HbA1c in the 2009 survey, and participants who received diabetes treatments. The ascertainment of T2DM according to high fasting plasma glucose/HbA1c in the 2009 survey was based on the American Diabetes Association criteria for the diagnosis of diabetes and the guideline for the prevention and treatment of Type 2 diabetes mellitus in China.
For dietary assessment, we calculated the cumulative average values of the participants from baseline to the last visit before the date of new-onset T2DM or the end of the study, to reflect the impact of long-term dietary habits on the risk of T2DM, as previous CHNS studies do [1, 2]. We have clarified this in more detail in the Method section, which can be found in the revised manuscript at Lines 141-142, Page 4 in red.
To explore the impact of the difference in FAs intake between dinner and breakfast on the risk of T2DM in people of different ages accordingly, we have conducted a subgroup analysis of age with 40 years old as the cutoff, since early-onset diabetes was defined as diabetes ascertained before 40 years old. The results showed that the positive association of Δ n-6/n-3 PUFAs between dinner and breakfast with T2DM risk were more pronounced among individuals aged > 40. Meanwhile, participants aged > 40 were found benefited more from Δ P-MUFA intake to against T2DM [3]. The added subgroup analysis can be found in the revised manuscript at Lines 314-320, Page 10 and Table S8 in red.
References:
- Ye, Z.; Wu, Q.; Yang, S.; Zhang, Y.; Zhou, C.; Liu, M.; Zhang, Z.; He, P.; Zhang, Y.; Li, R.; Li, H.; Liu, C.; Nie, J.; Hou, F.F.; Qin, X. Variety and quantity of dietary insoluble fiber intake from different sources and risk of new-onset hypertension. Bmc Med. 2023, 21, 61.
- Yan, M.; Liu, Y.; Wu, L.; Liu, H.; Wang, Y.; Chen, F.; Pei, L.; Zhao, Y.; Zeng, L.; Dang, S.; Yan, H.; Mi, B. The Association between Dietary Purine Intake and Mortality: Evidence from the CHNS Cohort Study. Nutrients. 2022, 14, 1718.
- Zeitler, P.; Galindo, R.J.; Davies, M.J.; Bergman, B.K.; Thieu, V.T.; Nicolay, C.; Allen, S.; Heine, R.J.; Lee, C.J. Early-Onset Type 2 Diabetes and Tirzepatide Treatment: A Post Hoc Analysis From the SURPASS Clinical Trial Program. Diabetes Care. 2024, 47, 1056-1064.
Comments 3: Since the reduction of the risk of T2D is between 19 and 29%, why the authors concluded that the timing is crucial?
Response 3: Thank you for your question. This study found that the highest category of Δ n-3 PUFAs and Δ P-MUFAs between dinner and breakfast were respectively linked to a 19% and 29% reduced T2DM risk, whild a higher n-6/n-3 ratio significantly increase 38% of the risk of T2DM. A recent CHNS study published in “Lancet Planet Health” reported that EAT-Lancet reference diet was related to a 25.3% decreased risk of T2DM [4]. And another study published in “Diabetes Care” conducted among 392,287 UK Biobank participants found that oily fish intakes were associated with 16-22% decreased risk of T2DM [5]. Our results are similar to these previous results on reducing the risk of T2DM, indicating that the timing of FAs intake play a crucial role in reducing the risk of T2DM. Based on Chrononutrition and considering the metabolic changes driven by the circadian rhythm, this study explored the temporal effect of FAs intakes on T2DM, rather than the absolute intake, providing new insights into the diet-specific FA timing for the prevention of T2DM.
References:
- Cai, H.; Talsma, E.F.; Chang, Z.; Wen, X.; Fan, S.; Van'T, V.P.; Biesbroek, S. Health outcomes, environmental impacts, and diet costs of adherence to the EAT-Lancet Diet in China in 1997-2015: a health and nutrition survey. Lancet Planet Health. 2024, 8, e1030-e1042.
- Chen, G.C.; Arthur, R.; Qin, L.Q.; Chen, L.H.; Mei, Z.; Zheng, Y.; Li, Y.; Wang, T.; Rohan, T.E.; Qi, Q. Association of Oily and Nonoily Fish Consumption and Fish Oil Supplements With Incident Type 2 Diabetes: A Large Population-Based Prospective Study. Diabetes Care. 2021, 44, 672-680.
Comments 4: I don't understand why the difference in the intake is used as an outcome and not the actual intake in each mealtime?
Response 4: Thank you for raising this critical methodological question. As we mentioned in the Introduction section, our approach was grounded in the chrononutrition hypothesis, which posits that the metabolic effects of nutrients depend not only on their quantity but also on their temporal distribution across circadian cycles [6, 7]. The Δ metric directly quantifies the “temporal redistribution of FAs between meals”, emphasizing the circadian rhythm’s role in modulating insulin sensitivity and glucose metabolism. We align with our objective to investigate how timing—rather than absolute intake—influences T2DM risk. Several previous studies investigated how diet timing influence health outcome, especially those related to circadian rhythms. For example, a recent study found that participants consume higher energy, fat and protein at dinner than breakfast were more likely to develop T2DM [8]. We have now clarified these methodological considerations in the revised manuscript, which can be found in the revised manuscript at Lines 66-84, Page 2 in red.
References:
- Henry, C.J.; Kaur, B.; Quek, R. Chrononutrition in the management of diabetes. Nutr. Diabetes. 2020, 10, 6.
- Flanagan, A.; Bechtold, D.A.; Pot, G.K.; Johnston, J.D. Chrono-nutrition: From molecular and neuronal mechanisms to human epidemiology and timed feeding patterns. J. Neurochem. 2021, 157, 53-72.
- Ren, X.; Yang, X.; Jiang, H.; Han, T.; Sun, C. The association of energy and macronutrient intake at dinner vs breakfast with the incidence of type 2 diabetes mellitus in a cohort study: The China Health and Nutrition Survey, 1997-2011. J. Diabetes. 2021, 13, 882-892.
Comments 5: Could you please explain the sources of omega-3 fatty acids in the studied population.
Response 5: We thank the reviewer's good suggestion. In CHNS, dietary data were collected by questionnaires of 3-d consecutive dietary intakes, and a 24-hour recall basis was applied to report all food consumed away from home during the face-to-face interview. All subjects were requested to continue their usual eating patterns. Intakes of various dietary fatty acids—SFAs, MUFAs, PUFAs, n-3 PUFAs, and n-6 PUFAs—were assessed by the fatty acid profiles of each food item and lipid conversion factors listed in the corresponding versions of Chinese Food Composition Table (FCT). All nutrient data were then expressed as a percentage of total energy intake (% kcal) via the energy density method. The main dietary sources of n-3 PUFAs include fish, shrimp, crab, shellfish and nuts. However, since the food items for calculating fatty acids are listed in versions of Chinese Food Composition Table (FCT; 1991, 2002, 2004 version), which was very large, we have cited the FCTs in our text at Line 123, Page 4 and Lines 521-526, Page 15. We have also listed some of the food source of FAs in the revised manuscript at Lines 131-142, Page 4 in red.
Comments 6: Finally, since dietary habits change during life due to several factors, are there fluctuations in fatty acids intake in the whole period studied
Response 6: Thank you for your question. To reduce the impact of fluctuations in FAs intake during the study period on the research results and to reflect the long-term dietary habits of the participants, we calculated the cumulative average FAs intake of the participants in each survey of this study till the onset of T2D or the end of the study, as previous studies do [9, 10]. We have clarified this in more detail in the Method section, which can be found in the revised manuscript at Lines 141-142, Page 4 in red.
References:
- Ye, Z.; Wu, Q.; Yang, S.; Zhang, Y.; Zhou, C.; Liu, M.; Zhang, Z.; He, P.; Zhang, Y.; Li, R.; Li, H.; Liu, C.; Nie, J.; Hou, F.F.; Qin, X. Variety and quantity of dietary insoluble fiber intake from different sources and risk of new-onset hypertension. Bmc Med. 2023, 21, 61.
- Yan, M.; Liu, Y.; Wu, L.; Liu, H.; Wang, Y.; Chen, F.; Pei, L.; Zhao, Y.; Zeng, L.; Dang, S.; Yan, H.; Mi, B. The Association between Dietary Purine Intake and Mortality: Evidence from the CHNS Cohort Study. Nutrients. 2022, 14, 1718.
Reviewer 3 Report
Comments and Suggestions for Authors
This research article examined the timing of dietary fatty acids to lower risk of T2DM based on using data from the China Health and Nutrition Survey. The objective of this study is very interesting. Several limitations should be addressed.
In title, it would be suggested to choose “China Health and Nutrition Survey” rather than “a Nationwide Longitudinal Chrononutrition Study”.
Authors need to write “type 2 diabetes mellitus (T2DM) rather than type 2 diabetes (T2D).
Authors need to clarify what SD stands for in the abstract and texts.
Introduction is too short. Introduction should provide sufficient evidence to scientifically support the purpose of the study.
Authors need to write more specifically in lines 60-66. In particular, it is better to address the contents of references 13 and 14 in detail.
Authors cited reference 18 in lines126- 127, “Previous study based on the 2009 survey had confirmed the accuracy of diagnosed diabetes in this cohort [18]. However, it is questionable whether the references cited here are appropriate references confirming diabetes diagnostic criteria.
Discussion is also short. Discussion should provide sufficient evidence to scientifically support the findings of this study with appropriate citations. Authors need to cite more relevant references with specific discussion.
Author Response
Comments 1: In title, it would be suggested to choose "China Health and Nutrition Survey" rather than "a Nationwide Longitudinal Chrononutrition Study".
Response 1: We appreciate the reviewer’s valuable suggestion and have revised the title, which can be found in the revised manuscript at Lines 2-4, Page 1 in red.
Comments 2: Authors need to write "type 2 diabetes mellitus (T2DM)" rather than "type 2 diabetes (T2D)".
Response 2: We appreciate the reviewer’s valuable suggestion and have changed the term "type 2 diabetes (T2D)" to "type 2 diabetes mellitus (T2DM)" throughout the manuscript in red.
Comments 3: Authors need to clarify what SD stands for in the abstract and texts.
Response 3: We apologize for not clarify the full name of SD, and we have added its full name "standard deviation" in the revised manuscript, which can be found at Line 34, Page 1 and Line 204, Page 5 in red.
Comments 4: Introduction is too short. Introduction should provide sufficient evidence to scientifically support the purpose of the study.
Response 4: We appreciate the reviewer’s valuable and have added more sufficient evidence to scientifically support the purpose of the study, including the heterogeneity of previous studies and the essential role of Chrononutrition in T2DM development. The total word count of Introduction section in the revised manuscript is 495 words. You can find the added contents at Lines 43-84, Page 2 in red.
Comments 5: Authors need to write more specifically in lines 60-66. In particular, it is better to address the contents of references 13 and 14 in detail.
Response 5: We appreciate the reviewer’s suggestion and have documented the heterogeneity of previous studies and Chrononutrition in more details. For example, our previous research showed that MUFA intake not significant associated with T2DM risk, while another study found that total MUFA intake might be essential in preventing prediabetes and T2DM. And Chrononutrition evidences indicate that aligning food intake to periods of the day when circadian rhythms in metabolic processes are optimised for nutrition may be effective for improving metabolic health. You can find the added contents in the revised manuscript at Lines 61-72, Page 2 in red.
Comments 6: Authors cited reference 18 in lines 126- 127, "Previous study based on the 2009 survey had confirmed the accuracy of diagnosed diabetes in this cohort [18]. However, it is questionable whether the references cited here are appropriate references confirming diabetes diagnostic criteria.
Response 6: Thank you for your question. The ascertainment of T2DM according to the level of fasting blood glucose or HbA1c was based on the American Diabetes Association criteria for the diagnosis of diabetes and the guideline for the prevention and treatment of Type 2 diabetes mellitus in China. And the definition of T2DM in previous CHNS studies is consistent with our study [1-3]. To demonstrate the scientific and accuracy of the T2DM definition, we have added several references, including several articles published in Diabetologia , Nutrients and Journal of Nutrition. You can find the added contents at Lines 153-161, Page 4 in red.
References:
- Zhuang, P.; Mao, L.; Wu, F.; Wang, J.; Jiao, J.; Zhang, Y. Cooking Oil Consumption Is Positively Associated with Risk of Type 2 Diabetes in a Chinese Nationwide Cohort Study. J. Nutr. 2020, 150, 1799-1807.
- Wang, H.; Gou, W.; Su, C.; Du W; Zhang, J.; Miao, Z.; Xiao, C.; Jiang, Z.; Wang, Z.; Fu, Y.; Jia, X.; Ouyang, Y.; Jiang, H.; Huang, F.; Li, L.; Zhang, B.; Zheng, J.S. Association of gut microbiota with glycaemic traits and incident type 2 diabetes, and modulation by habitual diet: a population-based longitudinal cohort study in Chinese adults. Diabetologia. 2022, 65, 1145-1156.
- Li, W.; Jiao, Y.; Wang, L.; Wang, S.; Hao, L.; Wang, Z.; Wang, H.; Zhang, B.; Ding, G.; Jiang, H. Association of Serum Magnesium with Insulin Resistance and Type 2 Diabetes among Adults in China. Nutrients. 2022, 14, 1799.
Comments 7: Discussion is also short. Discussion should provide sufficient evidence to scientifically support the findings of this study with appropriate citations. Authors need to cite more relevant references with specific discussion.
Response 7: We appreciate the reviewer’s valuable and have added more sufficient evidence to scientifically explain the results of the study, especially on how circadian rhythm system regulate the association of diet FAs and T2DM risk. To be more specific, circadian disruption and diabetes are reciprocally connected, and diet FAs can regulate hypothalamic and peripheral circadian clocks. The expression and activity glucokinase, a key enzyme for blood glucose metabolism, are regulated by circadian rhythm gene CLOCK and BMAL1, as well as by dietary signals. Meanwhile, n-3 PUFAs and fish oil were reported to beneficently affect the biological clock. However, the mechanism of functional fatty acids on circadian clock coordination and glycemic control remains unclear and further studies are needed. The total word count of Discussion section in the revised manuscript is 1389 words. You can find the added contents at Lines 321-429, Pages 10-12 in red.
Round 2
Reviewer 3 Report
Comments and Suggestions for Authors
This manuscript has been improved for the publication.
Author Response
Comment 1: This manuscript has been improved for the publication.
Response 1: Thank you.